# Time trend, social vulnerability, and identification of risk areas for tuberculosis in Brazil: An ecological study

João Paulo Silva de Paiva[1], Mônica Avelar Figueiredo Mafra Magalhães[2], Thiago Cavalcanti Leal[1], Leonardo Feitosa da Silva[1], Lucas Gomes da Silva[1], Rodrigo Feliciano do Carmo[3,4], Carlos Dornels Freire de Souza[1,5]*

1 Department of Medicine, Federal University of Alagoas, Arapiraca, AL, Brazil, 2 Health Information Laboratory, GIS Laboratory, Institute of Scientific and Technological Information and Communication in Health, Oswaldo Cruz Foundation, Rio de Janeiro, RJ, Brazil, 3 Postgraduate Program in Health and Biological Sciences, Federal University of Vale do São Francisco (UNIVASF), Petrolina, PE, Brazil, 4 Postgraduate Program in Biosciences, UNIVASF, Petrolina, PE, Brazil, 5 Postgraduate Program in Family Health, Department of Medicine, Federal University of Alagoas, Arapiraca, AL, Brazil

* carlos.freire@arapiraca.ufal.br

**Data Availability Statement:** This study did not require approval from the research ethics committee, because it used secondary data gathered from the public domain, where no

## Abstract

### Introduction

Tuberculosis is one of the ten leading causes of death and the leading infectious cause worldwide. The disease represents a challenge to health systems around the world. In 2018, it is estimated that 10 million people were affected by tuberculosis, and approximately 1.5 million people died due to the disease worldwide, including 251,000 patients coinfected with HIV. In Brazil, the disease caused 4,490 deaths, with rate of 2.2 deaths per 100,000 inhabitants. The objective of this study was to analyze the time behavior, spatial, spatial-temporal distribution, and the effects of social vulnerability on the incidence of TB in Brazil during the period from 2001 to 2017.

### Materials and methods

A spatial-temporal ecological study was conducted, including all new cases of tuberculosis registered in Brazil during the period from 2001 to 2017. The following variables were analyzed: incidence rate of tuberculosis, the Social Vulnerability Index, its subindices, its 16 indicators, and an additional 14 variables available on the Atlas of Social Vulnerability. The statistical treatment of the data consisted of the following three stages: a) time trend analysis with a joinpoint regression model; b) spatial analysis and identification of risk areas based on smoothing of the incidence rate by local empirical Bayesian model, application of global and local Moran statistics, and, finally, spatial-temporal scan statistics; and c) analysis of association between the incidence rate and the indicators of social vulnerability.

### Results

Brazil reduced the incidence of tuberculosis from 42.8 per 100,000 to 35.2 per 100,000 between 2001 and 2017. Only the state of Minas Gerais showed an increasing trend,

individual patients are identifiable. Public domain data from Brazilian government databases were used. raw data can be collected on the following platforms: https://datasus.saude.gov.br/acesso-a-informacao/casos-de-tuberculose-desde-2001-sinan/ https://datasus.saude.gov.br/populacao-residente http://ivs.ipea.gov.br/index.php/pt/.

**Funding:** The author(s) received no specific funding for this work.

**Competing interests:** The authors have declared that no competing interests exist.

whereas nine other states showed a stationary trend. A total of 326 Brazilian municipalities were classified as high priority, and 22 high-risk spatial-temporal clusters were identified. The overall Social Vulnerability Index and the subindices of Human Capital and Income and Work were associated with the incidence of tuberculosis. It was also observed that the incidence rates were greater in municipalities with greater social vulnerability.

## Conclusions

This study identified clusters with high risk of TB in Brazil. A significant association was observed between the incidence rate of TB and the indices of social vulnerability.

## Introduction

Tuberculosis (TB) is an infectious-contagious disease caused by *Mycobacterium tuberculosis*. TB is currently one of the ten leading causes of death and the leading infectious cause worldwide, ranking higher than HIV [1, 2]. TB affects both sexes and all age groups, although the male population over 15 years of age is at a higher risk [1].

The disease currently represents a challenge to health systems around the world [1–3]. In 2018 alone, it is estimated that 10 million people were affected by TB (5.7 million men, 3.2 million women, and 1.1 million children), and approximately 1.5 million people died due to TB worldwide, including 251,000 patients coinfected with HIV. Furthermore, of the 10 million patients, approximately 30% were not diagnosed and/or registered in official systems [1].

In 2018, in Brazil, 76,228 new cases of TB were notified, with an incidence of 36.6 cases per 100,000 inhabitants. Furthermore, the disease caused 4,490 deaths in Brazil, with a rate of 2.2 deaths per 100,000 inhabitants. In this continent-sized country, the distribution of TB follows a heterogeneous spatial pattern, with concentration in the axis that connects the North, Northeast, and Southeast Regions of the country [4, 5].

Concern regarding the incidence of TB led the World Health Organization (WHO) to implement the End TB Strategy in 2014. The objective of the strategy is to reduce the number of deaths due to TB by 95% and the incidence by 90% by 2035 [1]. For the period from 2016 to 2020, the WHO listed the following three groups of countries that are priority for the development of plans and strategies: i) elevated magnitude of TB, ii) TB associated with HIV, and iii) multidrug-resistant TB. With respect to Brazil, the country is part of two of these groups, ranking twentieth in magnitude of the disease and nineteenth in TB-HIV coinfection [6].

The 2019 global tuberculosis report indicates budgetary and technical difficulties in the current scenario, with a gap of more than four billion dollars for diagnosis, treatment, prevention, and research [1]. Nevertheless, it has already been observed that, for every dollar spent on reducing mortality due to TB, there is a return of up to 43 dollars; this demonstrates that, in addition to being necessary for public health and for reducing inequity, funds directed to the elimination of TB are also a sound investment [7].

Due to Brazil's situation within the global TB scenario, the Brazilian Ministry of Health has designed the National Plan for the End of Tuberculosis, with two fundamental goals: i) to reduce the incidence rate to less than 10 cases per 100,000 inhabitants and ii) to reduce the mortality rate of tuberculosis to less than 1 death per 100,000 inhabitants by the year 2035 [6].

In view of Brazil's commitment to fight to end TB and the audacious goals to be met, the monitoring of the disease incidence, the identification of risk areas, and the influence of social dynamics are elements that can contribute to controlling TB in Brazil and serve as a reference for other countries where the disease persists [1, 3, 8]. The success of the global strategy to end

TB depends on vigorous research capable of providing support to the decision-making process on the part of governments, in order to implement ambitious policies to fight the disease.

Based on this, the objective of this study was to analyze the time behavior, spatial, spatial-temporal distribution, and the effects of social vulnerability on the incidence of TB in Brazil during the period from 2001 to 2017.

## Materials and methods

### Ethical aspects

This study did not require approval from the research ethics committee, because it used secondary data gathered from the public domain, where no individual patients are identifiable.

### Study design

This is an ecological study including all new cases of TB registered in Brazil during the period from 2001 to 2017.

### Study area

Brazil is the fifth largest country by area in the world, with a territory of 8.5 million km$^2$. The country is divided into five major regions (North, Northeast, Central-West, Southeast, and South), comprising 26 states and the Federal District (**Fig 1**). The Brazilian population was estimated at 210 million inhabitants in 2018. The highest concentration is in the Southeast Region, with a population of 80 million inhabitants occupying 10.8% of the territory. At the other extreme, the North Region has approximately 16.3 million inhabitants in an area that occupies 45.2% of the national territory [9]. In addition to its continental size, the country is characterized by important socio-spatial inequalities. The Human Development Index (HDI) of the Southeast, Central-West, and South Regions is considered high (0.766, 0.757, and 0.754, respectively), while the HDI of the North and Northeast is classified as average (0.667 and 0.663, respectively) [10].

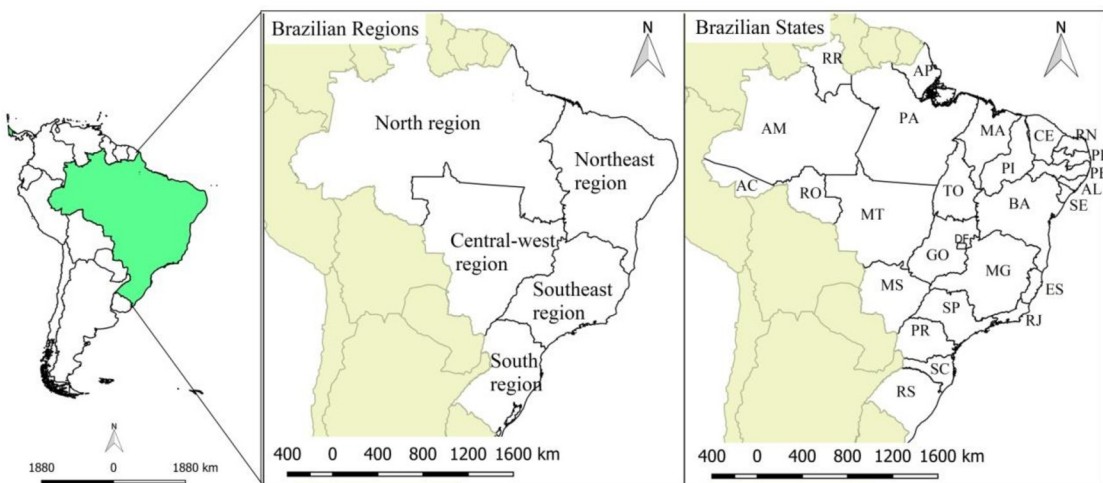

**Fig 1. Geographical location of the study area.** Map base layers were obtained from <http://www.naturalearthdata.com/about/terms-of-use/> covered by a Creative Commons Attribution 4.0 International (CCBY) License (https://creativecommons.org/licenses/by/4.0/legalcode). Map base layers were modified in QGIS software version 2.18.

## Data source

This study used the following databases:

1. Data referring to new cases of TB were extracted from the Brazilian Notifiable Diseases Information System (SINAN, acronym in Portuguese) (http://www2.datasus.gov.br). SINAN is a national coverage system that is responsible for the registration of all mandatory notification diseases throughout the country [11].

2. The population data necessary for calculating the incidence rate were obtained from the Brazilian Institute of Geography and Statistics, using the 2010 census and inter-census projections for the other years (http://www.ibge.gov.br).
The following equations were adopted:

*a) Annual incidence rate*

$$= \frac{(number\ of\ case\ registered\ as\ entry\ type\ 'new\ cases,\ unknown,\ and\ post-mortem'\ in\ the\ place\ and\ year)}{(resident\ population\ in\ the\ place\ and\ year)}\ x\ 100,000$$

*b) Average incidence rate of the period*

$$= \frac{(average\ cases\ registered\ as\ entry\ type\ 'new\ cases,\ unknown,\ and\ post-mortem'\ in\ the\ place)}{(resident\ population\ in\ the\ middle\ of\ the\ period)}\ x\ 100,000$$

3. We selected 34 social indicators that express living conditions and social vulnerability of the Brazilian population that may be associated with the maintenance of the transmission chain of tuberculosis in the Brazilian territory, as the literature has pointed out [12–14]. It should be noted that the indicators are available for all Brazilian municipalities (national coverage) and collected with technical quality by official institutions in the country.

Indicators of social vulnerability were obtained from the Social Vulnerability Atlas of the Brazilian Institute of Applied Economic Research (http://ivs.ipea.gov.br). The Social Vulnerability Index (SVI) was designed to indicate absent and/or insufficient living conditions throughout the territory of Brazil. The SVI varies from 0 to 1; the closer to 1, the greater the degree of social vulnerability. The classification is as follows: very low SVI (0 to 0.200), low SVI (0.201 to 0.300), medium SVI (0.301 to 0.400), high SVI (0.401 to 0.500), and very high SVI (0.501 a 1.000) [11].
The SVI is composed of 16 indicators, which are grouped into three subindices, as follows:

a. **Urban Infrastructure** (percentage of people in households with inadequate water supply and sanitary sewage; percentage of the population residing in urban households without trash collection service; percentage of people residing in households with per capita income less than half the minimum wage of 2010 who spend more than one hour to reach their workplace);

b. **Human Capital** (mortality up to 1 year of age; percentage of children between 0 and 5 years of age who do not attend school; percentage of children between 6 and 14 years of age who do not attend school; percentage of girls between 10 and 17 years of age who have had children; percentage of head-of-household mothers who have not completed primary school and who have at least 1 child under 15 years of age; illiteracy rate in the population age 15 years or over; percentage of children residing in households where none of the residents have completed primary school; percentage of people from 15 to 24 years of age who

do not study, do not work, and have a per capita household income less than or equal to half the minimum wage of 2010);

c. **Income and Work** (proportion of people with per capita household income less than or equal to half the minimum wage of 2010; unemployment rate of the population age 18 years or over; percentage of people age 18 years or older who have not completed primary schooling and who are informally employed; percentage of people in households with per capita income less than or equal to half the minimum wage of 2010 who are dependent on elderly individuals; activity rate of children between 10 and 14 years of age) [11].

In addition to these indicators, the following 14 variables available on the Social Vulnerability Atlas were included in this study: illiteracy rate (18 years or older); income per capita of people who are vulnerable to poverty; percentage of income from work; Gini index; percentage of employees with a formal contract (18 years or older); percentage of employees without a formal contract (18 years or older); percentage of public sector workers (18 years or older); percentage of self-employed workers (18 years or older); percentage of employers (18 years or older); degree of formality of employed people (18 years or older); percentage of employed people who have completed primary education (18 years or older); percentage of employed people who have completed higher education (18 years or older); average income of employed people (18 years or older); and percentage of employed people without income (18 years or older) [11].

## Data analysis

The statistical treatment of the data consisted of the following three stages:

**1) Time trend analysis.** For time trend analysis, the joinpoint regression model was used. This model tests whether a line with multiple segments is statistically better at describing the temporal evolution of a dataset, in comparison with a straight line or a line with fewer segments [15]. In this manner, the model makes it possible to identify an indicator's trend as stationary, increasing, or decreasing, as well as the points where there are changes in this trend (joins); this makes it possible to determine the annual percent change (APC) and the average annual percent change (AAPC).

The following parameters were adopted: minimum of 0 joins; maximum of 3 joins, and the model selection method was Monte Carlo permutation test with 4,499 permutations and autocorrelation of errors based on the data. A 95% confidence interval (95% CI) and a significance level of 5% were set. These analyses were carried out using the Joinpoint Regression Program (version 4.5.0.1, National Cancer Institute, Bethesda, MD, USA).

**2) Spatial analysis and identification of risk areas.** This analysis was subdivided into the following three stages: i) smoothing by local empirical Bayesian model, ii) global and local Moran statistics, and iii) spatial-temporal scan statistics.

Initially, the incidence rate of TB was corrected with the aid of a local empirical Bayesian model [16]. The objective of the modeling was to identify a posteriori distribution (unobserved quantities of a determined phenomenon), based on the application of Bayes' theorem, involving sample data (likelihood function), and a set of observed data (a priori distribution) [17]. This correction reduces random fluctuation caused by rare events, municipalities with small populations, and underreporting of the disease.

Having obtained the corrected coefficients, spatial autocorrelation was calculated using the global Moran index. The index provides a general measure of spatial association, whose expression for calculation considers an order 1 proximity matrix. The index ranges between −1 and +1. Values equal to zero indicate absence of spatial autocorrelation, and values close to

+1 and −1 indicate the existence of positive and negative spatial autocorrelation, respectively [18].

Once global dependence was established, the local Moran index (Local Index of Spatial Association [LISA]) was calculated. The spatial weight matrix adopted in this study was contiguity or adjacency matrix [Wij = 1], if [Ai] shares a common side with [Aj]; otherwise [Wij = 0], where [W] refers to Weight, and each value [Wij] depends on the spatial relationship between locations [i] and [j]. In these cases, municipalities share a border (first-order neighbors). All objects that are nearby were considered to have the same weight [18].

Based on the LISA, municipalities were positioned in quadrants of the Moran scatterplot as follows: Q1 (high-high), municipalities where the value of the attribute and the average value of the neighbors are higher than the average of the set, which are, thus, municipalities considered highest priority for intervention; Q2 (low-low), where the value of the attribute and the average of the neighbors are lower the average of the set; Q3 (high-low), where the value of the attribute is higher than that of the neighbors and the average of the neighbors is lower than that of the set; and Q4 (low-high), where the value of the attribute is lower than that of the neighbors and the average of the neighbors is higher than the average of the set. The municipalities classified as high-low and low-high have intermediate priority [18].

Spatial-temporal scan statistics were used to detect spatial-temporal clusters of municipalities with high risk of TB transmission. This statistical tool is based on the discrete Poisson probability model and the test to identify spatial-temporal clusters in the maximum likelihood method, whose alternative hypothesis is that there is a higher risk inside the window than outside [19]. The spatial-temporal scan is defined as a cylindrical window with a circular (or elliptical) geographic base, where the height corresponds to time. The base corresponds to purely spatial scan statistics and the height reflects the time period of potential clusters. Thus, the cylindrical window moves in space and time. Then an infinite number of overlapping cylinders of different shapes and sizes are obtained, in which each cylinder corresponds to a potential spatial cluster [19].

The scan statistic establishes a flexible circular window in the map, positioned on each of several centroids, whose radius is established in 50% of the total population at risk, with a maximum radius of 500 km. The flexibility of the window is justified by the fact that the size of the spatial-temporal clusters are not known *a priori*, given that the population at risk is not geographically homogeneous. Monte Carlo simulations (adopting 999 permutations) were used to obtain p values, and clusters with p value < 0.05 were considered significant. Subsequently, spatial-temporal clusters with two or fewer municipalities were excluded. Considering that the municipal spatial units are already delimited by objective criteria of the IBGE, in this work the edge effect was not considered in the results. During this stage, the software Terra View (version 4.2.2, Brazilian Space Research Institute, São José dos Campos, SP, Brazil), SatScan (version 9.1, National Cancer Institute, Bethesda, MD, USA), and QGis (version 2.14.11 Open Source Geospatial Foundation, Beaverton, OR, USA) were used.

**3) Analysis of association between the incidence rate and the indicators of social vulnerability.** Initially, bivariate spatial autocorrelation was used, with the TB incidence rate as the dependent variable and the social indicators studied as independent variables, identifying the global Moran index and the p value. Only variables with significant bivariate autocorrelation (p < 0.05) were used in the regression models. In order to avoid multicollinearity between variables, they were grouped into the following six blocks, according to specific themes:

Block 1- SVI;

Block 2- Subindices of the SVI;

Block 3- Urban Infrastructure variables of the SVI;

Block 4- Human Capital variables of the SVI;

Block 5- Income and Work variables of the SVI;

Block 6- Other indicators of social vulnerability.

For regression analysis, the decision model proposed by Luc Anselin was adopted [20]. According to this model, an ordinary least squares model is initially applied, followed by analysis of model residues by global Moran statistics. Once the global spatial dependence of these residues is established, one of the following two spatial regression models with global effects must be used: the spatial lag model (effects are noises that need to be removed) or the spatial error model (attributes the ignored spatial autocorrelation to the response variable Y). Robust Lagrange multiplier tests were used to determine which model to use [21].

The quality of the model was assessed by observing the Akaike (AIC), Schwarz Bayesian (BIC), $R^2$, log likelihood, and the Moran index statistic of the residues. The best model was the one whose AIC and BIC values were lower, the Log Likelihood and R2 were higher and the residuals showed spatial independence [21]. GeoDa software (version 1.10 Center for Spatial Data Science, Computation Institute, University of Chicago, Chicago, IL, USA) was used for this stage.

## Results

### Social vulnerability in Brazil

According to the SVI, Brazil was classified as a country with medium vulnerability, on both the overall index (SVI 0.326) and the Income and Work (SVI 0.320) and Human Capital (SVI 0.362) subindices. With respect to the Urban Infrastructure subindex, the country was classified as low vulnerability (SVI 0.295). The North Region had the highest vulnerability on the overall indicator (SVI 0.438) and on the Urban Infrastructure (SVI 0.419) and Human Capital (SVI 0.469) subindices, while the Northeast Region ranked highest on the Income and Work subindex (SVI 0.466) (**Fig 2**).

In 35.58% (n = 1,980) of Brazilian municipalities, overall social vulnerability was high or very high. It was also high or very high in 47.87% (n = 2,664) of municipalities on the Income and Work subindex; in 15.99% (n = 890) on the Urban Infrastructure index; and in 51.52% (n = 2,850) on the Human Capital subindex. The Income and Work and Human Capital subindices were the ones with the highest number of municipalities with very high social vulnerability (30.1% and 30.7%, respectively). The North and Northeast Regions showed more intense contexts of social vulnerability than the other regions.

Regarding the Urban Infrastructure subindex, a concentration of municipalities with high and very high vulnerability was observed in the North Region of the country (**Fig 2**).

### Tuberculosis: Magnitude, trend, and spatial-temporal risk clusters in Brazil

Between 2001 and 2017, 1,243,629 new cases of TB were registered in Brazil, with an incidence rate of 38.6 per 100,000. Throughout the time series, the regression model indicated a downward trend in the national rate (AAPC: −1.7; 95% CI: −2.0 to −1.4; p < 0.001), reducing from 42.8 per 100,000 (n = 73,800) to 35.2 per 100,000 (n = 69,626), between 2001 and 2017 (**Table 1**).

The highest incidence rates of the period were observed in the North (46.9 per 100,000) and in the Southeast (42.1 per 100,000). In terms of absolute number of new patients, the Southeast

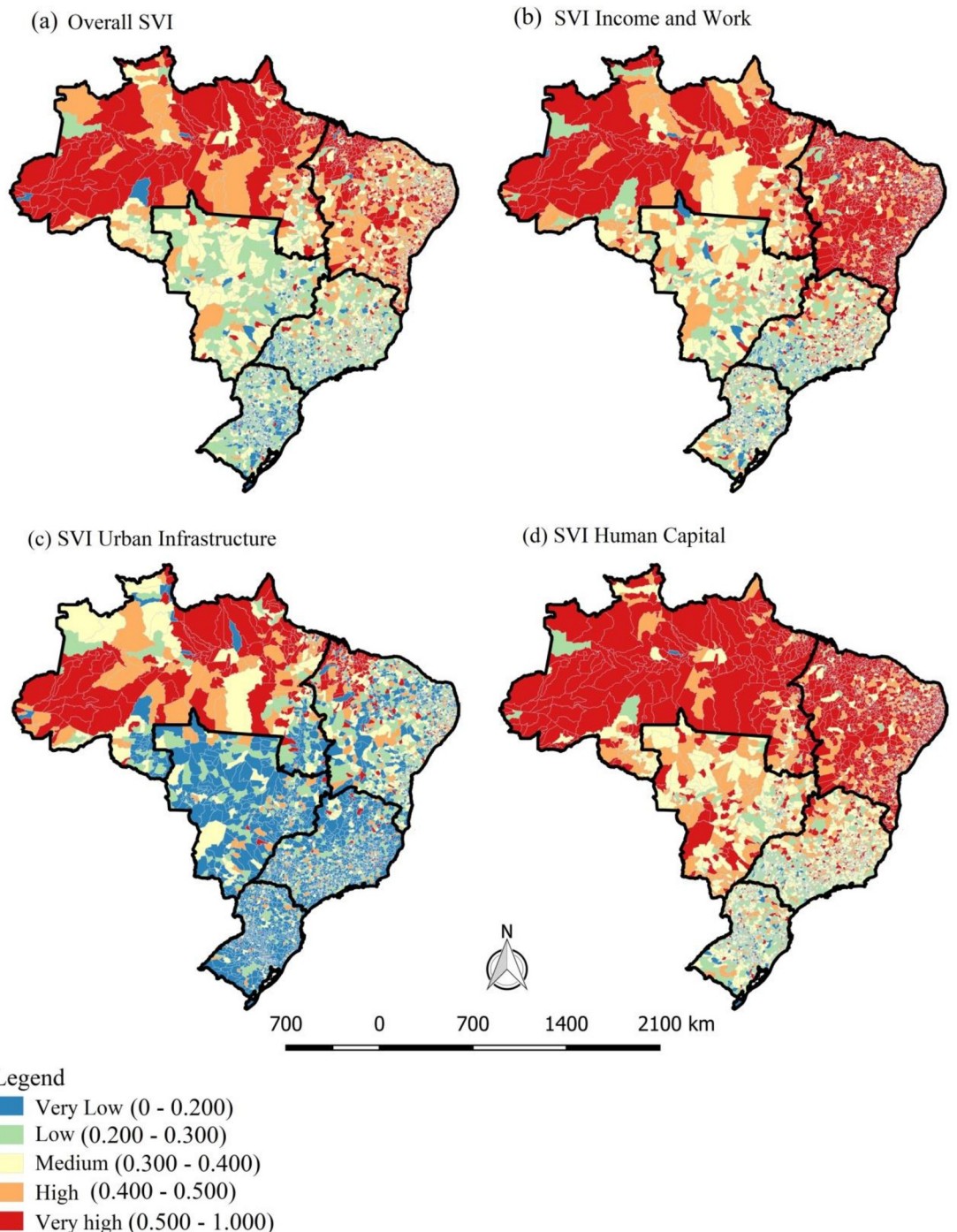

Fig 2. **Exploratory analysis of the Social Vulnerability Index in Brazil, 2010.** Map base layers were obtained from <http:// www.naturalearthdata.com/about/terms-of-use/> covered by a Creative Commons Attribution 4.0 International (CCBY) License (https://creativecommons.org/licenses/by/4.0/legalcode). Map base layers were modified in QGIS software version 2.18.

(45.4%; n = 563,200) and the Northeast (28.0%; n = 346,745) Regions ranked highest, accounting for more than two thirds of registrations. All five regions showed decreasing trends. The Northeast Region stood out with the greatest percentage decrease (AAPC: −2.3; 95% CI: −3.1 to −1.6; p < 0.001), whereas the North had the lowest decrease (AAPC: −0.8; 95% CI: −1.2 to

**Table 1. Trend of the incidence rate (100,000 inhabitants) of tuberculosis by Brazilian regions and states.** 2001–2017.

| Region/State | Incidence per 100,000 | | | Period | APC (95% CI) | AAPC (95% CI) |
|---|---|---|---|---|---|---|
| | **2001** | **2017** | **2001–2017** | | | |
| **Brazil** | 42.8 | 35.2 | 38.6 | 2001–2017 | −1.7* (−2.0; −1.4) | −1.7* (−2.0; −1.4) |
| **North** | 51.2 | 47.0 | 46.6 | 2001–2017 | −0.8* (−1.2; −0.4) | −0.8* (−1.2; −0.4) |
| Rondônia | 39.8 | 32.9 | 33.8 | 2001–2017 | −0.9 (−1.8; 0.1) | −0.9 (−1.8; 0.1) |
| Acre | 56.6 | 50.9 | 46.2 | 2001–2017 | −0.6 (−1.8; 0.6) | −0.6 (−1.8; 0.6) |
| Amazonas | 78.4 | 74.8 | 68.2 | 2001–2006 | −3.6* (−6.2; −1.0) | −0.3 (−1.2; 0.6) |
| | | | | 2006–2017 | 1.3* (0.5; 2.1) | |
| Roraima | 38.8 | 37.1 | 33.7 | 2001–2017 | −2.4* (−3.9; −0.8) | −2.4* (−3.9; −0.8) |
| Pará | 47.7 | 45.1 | 47.0 | 2001–2017 | −1.0* (−1.5; −0.6) | −1.0* (−1.5; −0.6) |
| Amapá | 38.9 | 31.1 | 33.9 | 2001–2015 | −4.4* (−5.8; −3.1) | −2.3 (−5.7; 1.3) |
| | | | | 2015–2017 | 14.2 (−15.8; 55.0) | |
| Tocantins | 22.6 | 10.1 | 15.0 | 2001–2017 | −4.6* (−5.2; −3.9) | −4.6* (−5.2; −3.9) |
| **Northeast** | 46.0 | 33.0 | 39.2 | 2001–2004 | −0.1 (−4.3; 4.6) | −2.3* (−3.1; −1.5) |
| | | | | 2004–2017 | −2.8* (−3.3; −2.4) | |
| Maranhão | 46.0 | 29.9 | 36.8 | 2001–2005 | −0.5 (−3.5; 2.6) | −2.6* (−3.7; −1.4) |
| | | | | 2005–2014 | −5.6* (−6.6; −4.6) | |
| | | | | 2014–2017 | 4.2 (−0.7; 9.3) | |
| Piauí | 40.7 | 20.2 | 37.4 | 2001–2008 | 7.1 (−0.7; 15.6) | −4.5* (−8.3; −0.6) |
| | | | | 2008–2017 | −12.7** (−17.1; −8.0) | |
| Ceará | 47.0 | 37.1 | 42.7 | 2001–2017 | −2.0* (−2.5; −1.5) | −2.0* (−2.5; −1.5) |
| Rio Grande do Norte | 37.0 | 31.0 | 32.1 | 2001–2017 | −2.1* (−2.8; −1.3) | −2.1* (−2.8; −1.3) |
| Paraíba | 32.8 | 26.2 | 29.3 | 2001–2017 | −1.5* (−2.2; −0.8) | −1.5* (−2.2; −0.8) |
| Pernambuco | 47.6 | 49.1 | 48.7 | 2001–2004 | 4.1 (−0.6; 9.0) | 0.0 (−1.7; 1.7) |
| | | | | 2004–2007 | −4.6 (−13.0; 4.7) | |
| | | | | 2007–2017 | 0.2 (−0.6; 0.9) | |
| Alagoas | 39.9 | 29.7 | 35.9 | 2001–2017 | −2.3* (−2.9; −1.7) | −2.3* (−2.9; −1.7) |
| Sergipe | 23.9 | 29.5 | 27.4 | 2001–2017 | 0.5 (−0.5; 1.5) | 0.5 (−0.5; 1.5) |
| Bahia | 55.4 | 28.3 | 39.3 | 2001–2017 | −4.2* (−4.6; −3.7) | −4.2* (−4.6; −3.7) |
| **Southeast** | 44.4 | 39.1 | 41.4 | 2001–2017 | −1.4* (−1.8; −1.0) | −1.4* (−1.8; −1.0) |
| Minas Gerais | 6.5 | 16.2 | 20.3 | 2001–2003 | 80.2* (58.5; 105.0) | 3.0* (1.5; 4.6) |
| | | | | 2003–2017 | −4.9* (−5.5; −4.3) | |
| Espírito Santo | 42.3 | 27.5 | 35.2 | 2001–2010 | −1.4 (−2.8; 0.1) | −2.7* (−3.8; −1.6) |
| | | | | 2010–2017 | −4.3* (−6.4; −2.2) | |
| Rio de Janeiro | 93.9 | 66.6 | 74.7 | 2001–2006 | −4.7* (−6.5; −2.9) | −2.6* (−3.2; −2.0) |
| | | | | 2006–2017 | −1.6* (−2.2; −1.1) | |
| São Paulo | 43.7 | 40.7 | 39.2 | 2001–2005 | −4.0* (−6.5; −1.5) | −0.7* (−1.4; −0.1) |
| | | | | 2005–2017 | 0.4 (−0.1; 0.9) | |
| **South** | 32.2 | 28.3 | 31.6 | 2001–2003 | 4.8* (0.6; 9.0) | −1.0* (−1.8; −0.2) |
| | | | | 2003–2006 | −5.1* (−8.8; −1.3) | |
| | | | | 2006–2011 | 1.6* (0.3; 2.9) | |
| | | | | 2011–2017 | −2.9* (−3.6; −2.3) | |
| Paraná | 27.2 | 18.0 | 23.1 | 2001–2017 | −2.8* (−3.1; −2.4) | −2.8* (−3.1; −2.4) |
| Santa Catarina | 24.8 | 25.2 | 26.7 | 2001–2013 | 0.6 (−0.2; 1.3) | 0.3 (−1.3; 0.8) |
| | | | | 2013–2017 | −2.8 (−6.6; 1.2) | |

(*Continued*)

**Table 1.** (Continued)

| Region/State | Incidence per 100,000 | | | Period | APC (95% CI) | AAPC (95% CI) |
|---|---|---|---|---|---|---|
| | 2001 | 2017 | 2001–2017 | | | |
| Rio Grande do Sul | 40.9 | 40.5 | 42.7 | 2001–2013 | 6.5 (−5.4; 19.8) | −0.3 (−3.0; 2.6) |
| | | | | 2003–2006 | −5.5 (−16.1; 6.3) | |
| | | | | 2006–2009 | 5.6 (−6.1; 18.9) | |
| | | | | 2009–2017 | −2.0* (−3.2; −0.7) | |
| **Central-West** | 28.7 | 21.1 | 23.9 | 2001–2009 | −3.1* (−3.9; −2.3) | −2.0* (−3.0; −0.9) |
| | | | | 2009–2013 | 2.8 (−1.1; 6.9) | |
| | | | | 2013–2017 | −4.3* (−6.7; −1.9) | |
| Mato Grosso do Sul | 39.7 | 33.1 | 36.0 | 2001–2017 | −1.2* (−1.8; −0.6) | −1.2* (−1.8; −0.6) |
| Mato Grosso | 47.5 | 34.6 | 39.4 | 2001–2009 | −3.3* (−6.0; −0.6) | −1.8 (−5.1; 1.7) |
| | | | | 2009–2013 | 9.1 (−4.0; 23.9) | |
| | | | | 2013–2017 | −8.7* (−15.8; −1.0) | |
| Goiás | 19.8 | 14.5 | 15.6 | 2001–2006 | −6.5* (−10.1; −2.7) | −2.5* (−3.8; −1.2) |
| | | | | 2006–2017 | −0.6 (−1.8; 0.6) | |
| Federal District | 16.4 | 10.2 | 13.7 | 2001–2017 | −2.8* (−3.8; −1.7) | −2.8* (−3.8; −1.7) |

*p < 0.05; APC: annual percent change; AAPC: average annual percent change; CI: confidence interval. Minimum joins: 0; maximum joins: 3; model selection method: permutation test with 4,499 permutations. Significance of 95%. Autocorrelation of errors based on the data.

−0.5; p < 0.001). The incidence rate in the South Region showed the greatest instability, with four time trends, interspersing periods of increase and decline (**Table 1**).

In terms of states, the highest rates were observed in Rio de Janeiro (74.7 per 100,000), Amazonas (68.2 per 100,000), and Pernambuco (48.7 per 100,000). These states accounted for 25.07% (n = 311,660) of all cases registered during the period. The time trend was decreasing in 16 states and in the Federal District, as follows: three in the North Region, seven in the Northeast, three in the Southeast, one in the South, and three in the Central-West (two states and the Federal District). Minas Gerais was the only state with an increasing trend throughout the period studied (APC: 3.0%; 95% CI: 1.5 to 4.6). The remaining states showed a stationary pattern (**Table 1**).

Regarding municipalities, 22.6% (n = 1,261) registered fewer than 10 cases of TB during the period studied, and only 5.6% (n = 311) registered fewer than 500 cases. The Moran statistics showed spatial dependence in both the crude rates (I 0.344; p = 0.01) and the rates smoothed by the local empirical Bayesian model (I 0.511; p = 0.01). Regarding the crude rates, 21.85% (n = 1,216) of municipalities showed incidence rates of ≤ 10.0 per 100,000, and 9.18% (n = 511) showed rates above 40.0 per 100,000. After smoothing, a reduction was observed in the proportion of municipalities with incidence rates of ≤ 10.0 per 100,000, which changed to 5.48% (n = 305). On the Moran map, 14.14% (n = 787) of municipalities were located in the Q1 quadrant of the Moran scatterplot (high-high) and were considered priority for intervention, as follows: 326 in the Northeast, 113 in the North, 55 in the Central-West, 208 in the Southeast, and 85 in the South (**Fig 3**).

The spatial-temporal scan statistics identified 22 spatial-temporal clusters with high risk of transmission of TB in Brazil, involving 561 municipalities, a number of municipalities smaller than that observed in Moran's statistics (787 vs 561; 71.28%). Fifteen spatial-temporal clusters were located in the Northeast and Southeast Regions, with six in the state of São Paulo. Clusters 9, 11, and 7, located in São Paulo, showed the highest relative risks (RR 6.49; RR 4.70, and RR 3.76, respectively). Cluster 8 concentrated the highest the highest number of locations, namely, 162 municipalities in the states of São Paulo, Santa Catarina, and Paraná, with an

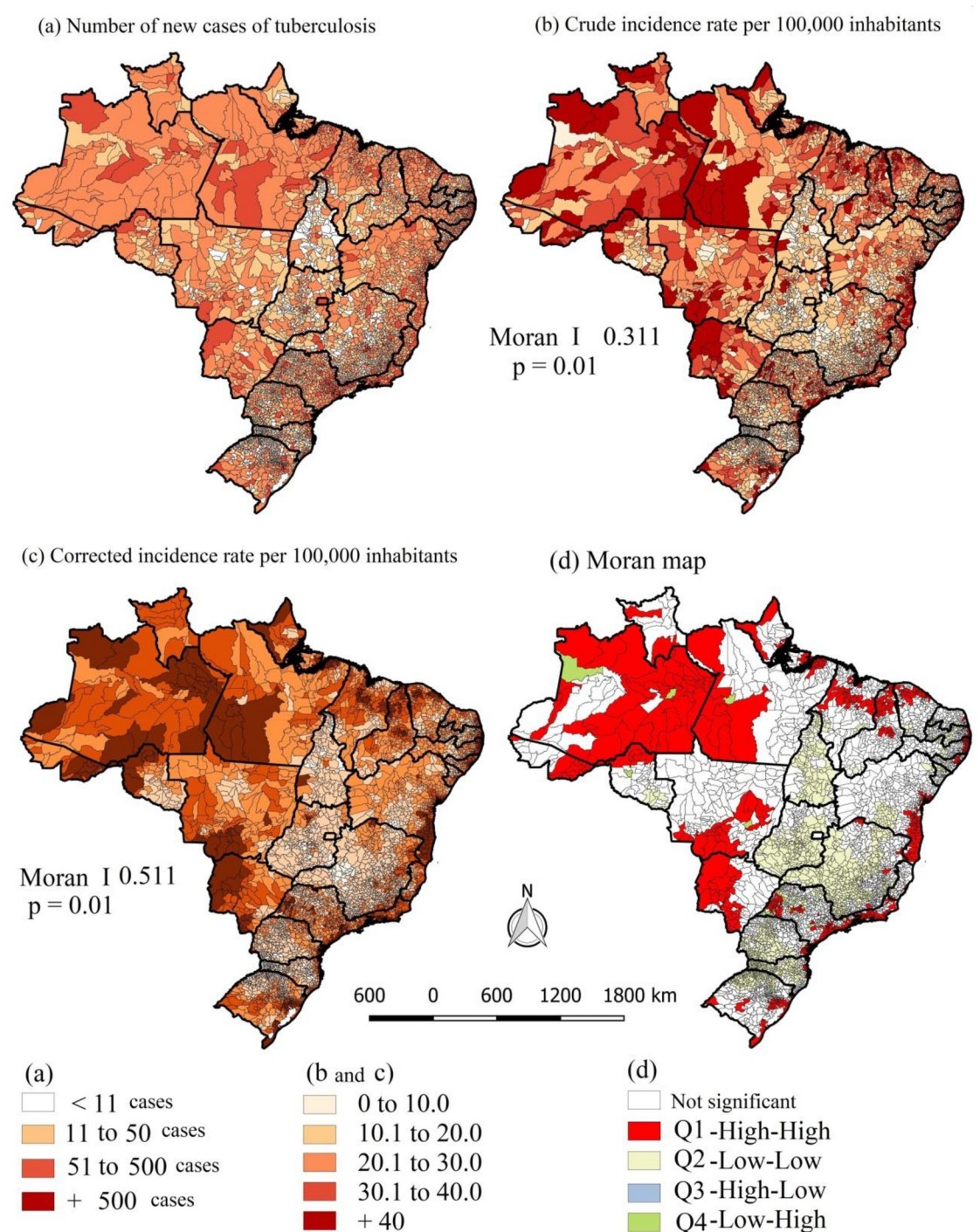

**Fig 3. Spatial distribution of incidence rates of tuberculosis in Brazil, crude and smoothed by the local empirical Bayesian model, and Moran Map, 2001–2017.** Map base layers were obtained from <http://www.naturalearthdata.com/about/terms-of-use/> covered by a Creative Commons Attribution 4.0 International (CCBY) License (https://creativecommons.org/licenses/by/4.0/legalcode). Map base layers were modified in QGIS software version 2.18.

incidence rate of 55.6 per 100,000 and RR 1.21. In the Northeast, cluster 10 stood out, with 117 municipalities in the states of Maranhão and Piauí, with an incidence rate of 60.3 per 100,000 and RR 1.31 (**Table 2** and **Fig 4**).

**Table 2. Spatial-temporal clusters of tuberculosis.** Brazil, 2001–2017.

| Cluster | Period | Region | Radius (km) | Number of municipalities | Number of cases | Rate per 100,000 | Relative risk | p value |
|---------|--------|--------|-------------|--------------------------|-----------------|------------------|---------------|---------|
| 1 | 2001–2008 | Rio de Janeiro | 64.88 | 25 | 108,696 | 117.7 | 2.68 | <0.001 |
| 2 | 2008–2015 | Rio Grande do Sul | 64.82 | 22 | 27,790 | 120.5 | 2.65 | <0.001 |
| 3 | 2010–2017 | Pernambuco | 27.90 | 10 | 29,124 | 104.6 | 2.30 | <0.001 |
| 4 | 2001–2008 | Bahia | 30.81 | 6 | 24,580 | 110.4 | 2.42 | <0.001 |
| 5 | 2010–2017 | Pará | 28.34 | 6 | 18,257 | 104.7 | 2.29 | <0.001 |
| 6 | 2003–2010 | Ceará | 22.71 | 6 | 21,016 | 86.1 | 1.88 | <0.001 |
| 7 | 2010–2017 | São Paulo | 48.66 | 25 | 3,989 | 173.0 | 3.76 | <0.001 |
| 8 | 2001–2008 | São Paulo/Paraná/Santa Catarina | 215.65 | 162 | 107,675 | 55.6 | 1.22 | <0.001 |
| 9 | 2010–2017 | São Paulo | 15.80 | 4 | 906 | 6.49 | 6.49 | <0.001 |
| 10 | 2001–2008 | Maranhão | 198.75 | 117 | 21,952 | 60.3 | 1.31 | <0.001 |
| 11 | 2010–2017 | São Paulo | 26.52 | 4 | 454 | 96.58 | 4.70 | <0.001 |
| 12 | 2002–2009 | Minas Gerais/Bahia/ Espírito Santo | 110.66 | 39 | 4,302 | 62.8 | 1.37 | <0.001 |
| 13 | 2011–2017 | Acre | 131.10 | 6 | 2,179 | 69.5 | 1.51 | <0.001 |
| 14 | 2008–2014 | Santa Catarina | 89.73 | 49 | 7,677 | 56.4 | 1.23 | <0.001 |
| 15 | 2012–2017 | São Paulo | 37.74 | 12 | 756 | 88.8 | 1.93 | <0.001 |
| 16 | 2001–2008 | Mato Grosso do Sul | 154.91 | 10 | 1,560 | 71.1 | 1.54 | <0.001 |
| 17 | 2006–2009 | Piauí | 65.41 | 25 | 805 | 83.4 | 1.81 | <0.001 |
| 18 | 2001–2008 | Pará/Amazonas/Mato Grosso | 237.06 | 5 | 1,060 | 70.1 | 1.52 | <0.001 |
| 19 | 2003–2004 | Minas Gerais | 55.08 | 5 | 142 | 113.2 | 2.46 | <0.001 |
| 20 | 2013–2017 | São Paulo | 35.33 | 9 | 1,452 | 57.7 | 1.25 | <0.001 |
| 21 | 2001–2005 | Rondônia | 107.04 | 4 | 302 | 75.4 | 1.64 | <0.001 |
| 22 | 2006–2006 | Piauí | 93.65 | 10 | 126 | 90.5 | 1.96 | <0.001 |

When comparing the Moran statistics and the spatial-temporal scan, areas of disagreement were observed in the presence of spatial clusters: while in the Moran statistics, areas of risk appear in the states of Amazonas, Roraima and Amapá (in the North), in Mato Grosso (Central-West) and Rio Grande do Norte, Paraíba and Alagoas (in the Northeast), these states do not have risk clusters in the spatial-temporal analysis (Figs 3 and 4).

## Association between social vulnerability and the incidence of tuberculosis in Brazil

Only the variable "percentage of people age 18 years or older who have not completed primary schooling and who are informally employed" did not show bivariate spatial correlation and was excluded from the regression model. In analysis of the six blocks of variables, there was spatial dependence of the residues of the ordinary least squares model, and the Lagrange multiplier tests indicated spatial lag as the spatial model to be applied.

The overall SVI and the Human Capital and Income and Work subindices were associated with the incidence of tuberculosis. In the disaggregate analysis of variables, association was observed with one variable from the Urban Infrastructure subindex (percentage of people residing in households with per capita income less than half the minimum wage of 2010 who spend more than one hour to reach their workplace), six from the Human Capital subindex (percentage of children between 6 and 14 years of age who do not attend school; percentage of girls between 10 and 17 years of age who have had children; percentage of head-of-household mothers who have not completed primary school and who have at least 1 child under 15 years of age; illiteracy rate in the population age 15 years or over; percentage of children residing in households where none of the residents have completed primary school; and percentage of

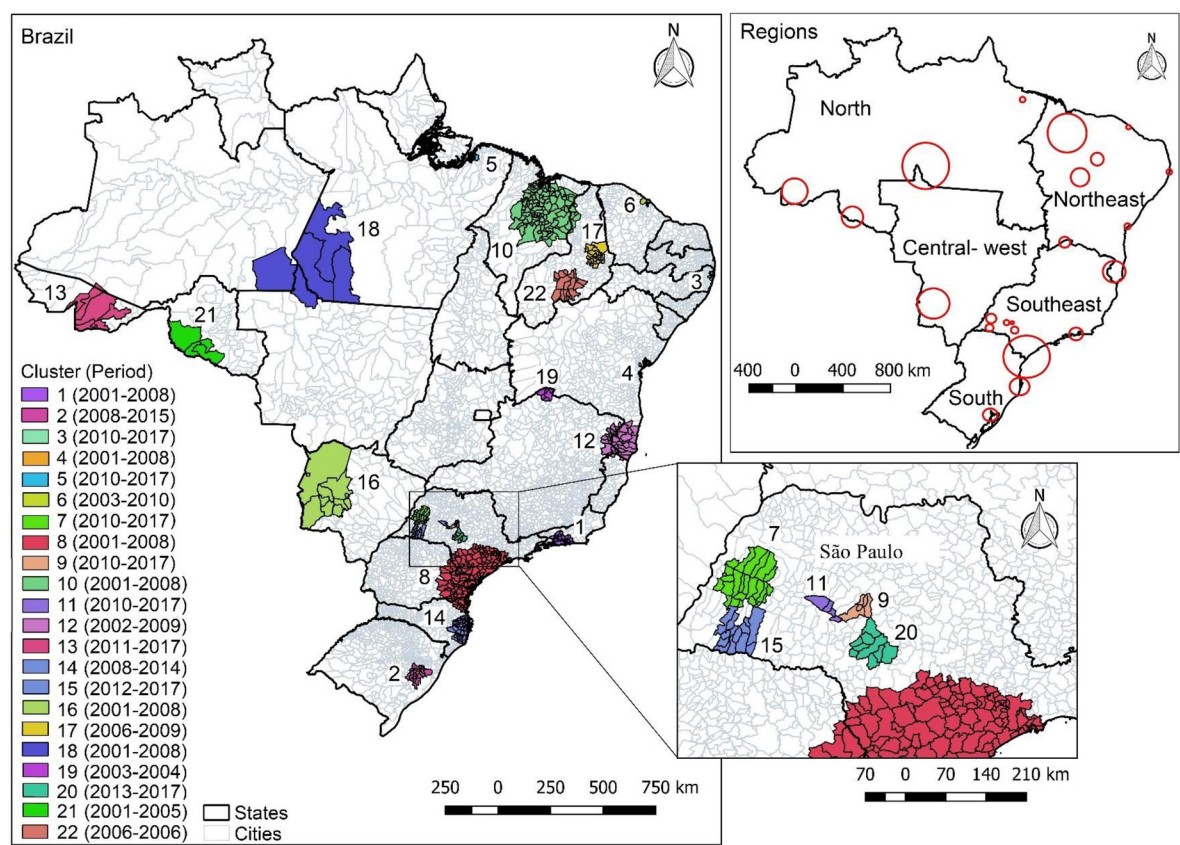

**Fig 4. Spatial-temporal scan statistics of cases of tuberculosis.** Brazil, 2001–2017. Map base layers were obtained from <http://www.naturalearthdata.com/about/terms-of-use/> covered by a Creative Commons Attribution 4.0 International (CCBY) License (https://creativecommons.org/licenses/by/4.0/legalcode). Map base layers were modified in QGIS software version 2.18.

people from 15 to 24 years of age who do not study, do not work, and have a per capita household income less than or equal to half the minimum wage of 2010), three variables from the Income and Work subindex (proportion of people with per capita household income less than or equal to half the minimum wage of 2010; unemployment rate of the population age 18 years or over; and percentage of people in households with per capita income less than or equal to half the minimum wage of 2010 who are dependent on elderly individuals), and seven additional variables that also express vulnerability, but that were not contemplated in the SVI (illiteracy rate [18 years of age or older]; income per capita of people who are vulnerable to poverty; percentage of income from work; percentage of employers [18 years of age or older]; percentage of employed people who have completed primary education [18 years of age or older]; percentage of employed people who have completed higher education [18 years of age or older]; and average income of employed people [18 years of age or older]) (**Table 3**). In all analyses, the Spatial Lag Model performed better when compared to the OLS model (**S1 Table**).

## Effects of social vulnerability on the incidence trend of tuberculosis in Brazil

Incidence rates were higher in municipalities with greater social vulnerability. The incidence rate was 49.2% higher in municipalities with very high overall SVI when compared to the

**Table 3. Bivariate spatial autocorrelation and classical and spatial regression models of factors associated with the incidence rate of tuberculosis, corrected by the local empirical Bayesian model.** Brazil, 2001–2017.

| Indicators of social vulnerability | Bivariate spatial autocorrelation | | Classical and spatial regression models | | | |
|---|---|---|---|---|---|---|
| | I Moran | p value | Ordinary least squares- OLS | | Spatial lag model- SLM | |
| | | | Rate | p value | Rate | p value |
| **Block 1- Social Vulnerability Index** | | | | | | |
| Constant | – | – | 20.401 | <0.001 | 4.141 | <0.001 |
| Social Vulnerability Index | 0.1572 | 0.001 | 25.362 | <0.001 | 4.946 | 0.006 |
| **Block 2- Subindices of the Social Vulnerability Index** | | | | | | |
| Constant | – | – | 21.786 | <0.001 | 5.058 | <0.001 |
| SVI Urban Infrastructure | 0.0074 | 0.045 | −0013 | 0.890 | −0.004 | 0.591 |
| SVI Human Capital | 0.1374 | 0.001 | 40.111 | <0.001 | 10.182 | 0.001 |
| SVI Income and Work | 0.0838 | 0.001 | −23.104 | <0.001 | −8.636 | 0.001 |
| **Block 3- Variables of SVI Urban Infrastructure** | | | | | | |
| Constant | – | – | 23.411 | <0.001 | 4.082 | <0.001 |
| % of people in households with inadequate water supply and sanitary sewage | 0.1245 | 0.001 | 0.098 | <0.001 | −0.026 | 0.187 |
| % of the population residing in urban households without trash collection service | 0.0990 | 0.001 | 0.0883 | <0.001 | 0.013 | 0.575 |
| % of people residing in households with per capita income less than half the minimum wage of 2010 who spend more than one hour to reach their workplace | 0.1337 | 0.001 | 0.718 | <0.001 | 0.363 | <0.01 |
| **Block 4- Variables of SVI Human Capital** | | | | | | |
| Constant | – | – | 30.524 | <0.001 | 8.574 | <0.001 |
| Mortality up to 1 year of age | 0.1224 | 0.001 | 0.4060 | <0.001 | 0.0842 | 0.165 |
| % of children between 0 and 5 years of age who do not attend school | −0.0472 | 0.001 | −0.103 | <0.001 | −0.017 | 0.446 |
| % of children between 6 and 14 years of age who do not attend school | 0.1257 | 0.001 | 0.945 | <0.001 | 0.342 | <0.001 |
| % percentage of girls between 10 and 17 years of age who have had children | 0.1210 | 0.001 | 0.763 | <0.001 | 0.332 | 0.016 |
| % of head-of-household mothers who have not completed primary school and who have at least 1 child under 15 years of age | 0.1296 | 0.001 | 0.215 | <0.001 | 0.082 | 0.005 |
| Illiteracy rate in the population age 15 years or over | 0.0674 | 0.001 | −0.411 | <0.001 | −0.161 | 0.004 |
| % of children residing in households where none of the residents have completed primary school | 0.0269 | 0.001 | −0.332 | <0.001 | −0.172 | <0.001 |
| % percentage of people from 15 to 24 years of age who do not study, do not work, and have a per capita household income less than or equal to half the minimum wage of 2010 | 0.1372 | 0.001 | 0.506 | <0.001 | 0.227 | <0.001 |
| **Block 5- Variables of SVI Income and Work** | | | | | | |
| Constant | – | – | 38.985 | <0.001 | 12.845 | <0.001 |
| Proportion of people with per capita household income less than or equal to half the minimum wage of 2010 | 0.1139 | 0.001 | 0.279 | <0.001 | 0.086 | 0.002 |
| Unemployment rate of the population age 18 years or over | 0.2158 | 0.001 | 1.007 | <0.001 | 0.561 | <0.001 |
| % of people age 18 years or older who have not completed primary schooling and who are informally employed | −0.0021 | 0.2630 | – | – | – | – |
| % of people in households with per capita income less than or equal to half the minimum wage of 2010 who are dependent on elderly individuals | 0.1086 | 0.001 | 0.289 | 0.187 | 0.190 | <0.001 |
| Activity rate of children between 10 and 14 years of age | −0.1482 | 0.001 | −0.197 | <0.001 | −0.055 | 0.097 |
| **Block 6- Other indicators of social vulnerability** | | | | | | |
| Constant | – | – | 47.562 | 0.003 | 19.292 | 0.165 |
| Illiteracy rate (18 years or older) | 0.0674 | 0.001 | 1.645 | 0.004 | 1.319 | 0.007 |
| Income per capita of people who are vulnerable to poverty | −0.1052 | 0.001 | −0.289 | <0.001 | −0.133 | <0.001 |
| % of income from work | −0.0346 | 0.001 | −0.224 | <0.001 | −0.124 | 0.002 |
| Gini index | 0.1068 | 0.001 | 16.765 | 0.025 | 4.135 | 0.514 |
| % of employees with a formal contract (18 years or older) | 0.0446 | 0.001 | 0.458 | 0.004 | 0.143 | 0.286 |
| % of employees without a formal contract (18 years or older) | 0.0283 | 0.001 | 0.128 | 0.352 | 0.118 | 0.313 |
| % of public sector workers (18 years or older) | −0.6841 | 0.001 | 0.231 | 0.181 | 0.151 | 0.302 |

*(Continued)*

**Table 3.** (Continued)

| Indicators of social vulnerability | Bivariate spatial autocorrelation | | Classical and spatial regression models | | | |
|---|---|---|---|---|---|---|
| | I Moran | p value | Ordinary least squares- OLS | | Spatial lag model- SLM | |
| | | | Rate | p value | Rate | p value |
| % of self-employed workers (18 years or older) | −0.0562 | 0.001 | 0.138 | 0.341 | 0.015 | 0.897 |
| % of employers (18 years or older) | −0.0411 | 0.001 | −0.687 | 0.052 | −0.668 | 0.026 |
| Degree of formality of employed people (18 years or older) | −0.011 | 0.001 | −0.229 | <0.001 | −0.065 | 0.208 |
| % of employed people who have completed primary education (18 years or older) | 0.0545 | 0.001 | 0.460 | <0.001 | 0.143 | 0.102 |
| % of employed people who have completed higher education (18 years or older) | −0.0221 | 0.001 | −1.511 | <0.001 | −0.884 | <0.001 |
| Average income of employed people (18 years or older) | −0.0197 | 0.001 | 0.006 | 0.009 | 0.005 | 0.010 |
| % percentage of employed people without income (18 years or older) | −0.0134 | 0.001 | −0.343 | 0.013 | −0.151 | 0.197 |

group with very low SVI (29.32 per 100,000 and 19.64 per 100,000, respectively). On the Urban Infrastructure subindex, this percentage was 37.2% (31.02 per 100,000 where SVI was very high and 22.60 per 100,000 where SVI was very low); on Income and Work, it was 17.1% (28.16 per 100,000 and 24.04 per 100,000, respectively), and, on Human Capital, it was 74.2% (29.22 per 100,000 and 16.77 per 100,000, respectively) (**Table 4**).

Only the municipalities classified as high vulnerability on the Urban Infrastructure subindex and those classified as very low vulnerability on the Human Capital subindex showed an increasing trend in incidence throughout the time series (APC 0.3% and 1.4%, respectively). A decreasing trend was observed only in municipalities with very high social vulnerability for Income and Work (APC −1.8%). In groups of municipalities with very high vulnerability, a decreasing trend was found in incidence until the year 2015, when there was an interruption in this trend. For all the strata of social vulnerability, the incidence rates in 2016 and 2017 were greater than those observed in 2015 (**Table 4**, **S2 Table**).

## Discussion

In spite of all the global efforts in the fight against TB and the successive plans to eliminate the disease, it still represents one of the most important challenges for public health, especially in low- and middle-income countries. In these places, the disease burden is elevated, and the reduction in incidence and mortality tends to occur more slowly [1, 2].

Although Brazil has played a prominent role in the global fight against TB [13], the national scenario is still not favorable to the fulfillment of the goals put forth in the global strategy and the national plan for ending TB. This is evidenced by the slow annual reduction in incidence rate observed in this study (−1.7%) and by the increase in the rate in 2016 and 2017, which suggests a possible reversal of this trend in the coming years. This scenario suggests that reaching the goals for 2035 will demand intense effort on the part of Brazil.

In the same manner, the world is not on the right track to reach the agreed milestones (20% by 2020, 50% by 2025, 80% by 2030, and 90% by 2035) for the incidence rate of TB. Even with worldwide effort, the current speed of reduction in the global incidence is insufficient to meet the goals for eliminating TB. Globally speaking, the average rate of decline in the incidence rate of TB was 1.6% per year during the period from 2000 to 2018 and 2.0% between 2017 and 2018 [22]. The accumulated reduction between 2015 and 2018 was merely 6.3% [22].

In Brazil, the challenges imposed on TB control are even greater, given that they involve a wide and complex spectrum of social determinants of health [13]. In a continent-sized country, regional socioeconomic inequalities and different contexts of social vulnerability

**Table 4. Analysis of the incidence trend of tuberculosis per 100,000 inhabitants by Social Vulnerability Index classification.** Brazil, 2001–2017.

| Overall SVI | Number of municipalities | Incidence per 100,000 | Period | APC | (95% CI) | Classification |
|---|---|---|---|---|---|---|
| Very low | 627 | 19.64 | 2001–2017 | −0.01 | −0.9; 0.7 | Stationary |
| Low | 1,699 | 24.03 | 2001–2017 | −0.3 | −1.0; 0.4 | Stationary |
| Medium | 1,258 | 27.48 | 2001–2003 | 13.5 | 3.1; 25.0 | Increasing |
| | | | 2003–2015 | −2.8 | −2.8; −2.2 | Decreasing |
| | | | 2015–2017 | 4.9 | 4.9; −4.7 | Stationary |
| | | | 2001–2017 | 4.9 | −4.7; 15.5 | Stationary |
| High | 1,178 | 29.29 | 2001–2003 | 13.3 | 0.1; 28.2 | Increasing |
| | | | 2003–2017 | −3.4 | −3.4; −3.9 | Decreasing |
| | | | 2001–2017 | −1.4 | −1.4; −2.9 | Stationary |
| Very high | 802 | 29.32 | 2001–2005 | 4.3 | −0.1; 9.0 | Stationary |
| | | | 2005–2015 | −5.1 | −5.1; −3.9 | Decreasing |
| | | | 2015–2017 | 4.2 | 4.2; −9.2 | Stationary |
| | | | 2001–2017 | −1.7 | −1.7; −3.5 | Stationary |
| **SVI Urban Infrastructure** | **Number of municipalities** | | **Period** | **APC** | **(95% CI)** | **Classification** |
| Very low | 2,815 | 22.60 | 2001–2003 | 13.5 | 7.8; 19.5 | Increasing |
| | | | 2003–2015 | −2.7 | −3.0; −2.3 | Decreasing |
| | | | 2015–2017 | 4.7 | −0.6; 10.3 | Stationary |
| | | | 2001–2017 | 0.1 | −0.7; 1.0 | Stationary |
| Low | 1,100 | 28.10 | 2001–2003 | 10.0 | −2.2; 23.7 | Stationary |
| | | | 2003–2017 | −2.3 | −2.3; −2.8 | Decreasing |
| | | | 2001–2017 | −0.8 | −0.8; −2.2 | Stationary |
| Medium | 758 | 28.87 | 2001–2003 | 12.1 | −2.2; 28.6 | Stationary |
| | | | 2003–2017 | −2.5 | −3.1; −1.9 | Decreasing |
| | | | 2001–2017 | −0.8 | −2.4; 0.9 | Stationary |
| High | 518 | 34.24 | 2001–2003 | 8.0 | −1.2; 18.2 | Stationary |
| | | | 2003–2015 | −1.8 | −2.4; −1.3 | Decreasing |
| | | | 2015–2017 | 6.1 | 6.1; −3.0 | Stationary |
| | | | 2001–2017 | 0.3 | −1.1; 1.8 | Increasing |
| Very high | 372 | 31.02 | 2001–2003 | 16.8 | 2.6; 32.9 | Increasing |
| | | | 2003–2015 | −4.3 | −5.1; −3.4 | Decreasing |
| | | | 2015–2017 | 4.8 | −7.9; 19.3 | Stationary |
| | | | 2001–2017 | −0.7 | −2.8; 1.3 | Stationary |
| **SVI Income and Work** | **Number of municipalities** | | **Period** | **APC** | **(95% CI)** | **Classification** |
| Very low | 335 | 24.04 | 2001–2017 | −0.4 | −1.0; 0.2 | Stationary |
| Low | 1,317 | 25.56 | 2001–2017 | 0.1 | −0.6; 0.8 | Stationary |
| Medium | 1,248 | 24.51 | 2001–2003 | 11.1 | 2.8; 26.8 | Stationary |
| | | | 2003–2017 | −1.2 | −1.9; −0.6 | Decreasing |
| | | | 2001–2017 | 0.2 | −1.4; 1.8 | Stationary |
| High | 987 | 26.54 | 2001–2003 | 14.4 | −2.4; 34.0 | Stationary |
| | | | 2003–2017 | −3.3 | −4.0; −2.5 | Decreasing |
| | | | 2001–2017 | −1.2 | −1.2; −3.0 | Stationary |
| Very high | 1,677 | 28.16 | 2001–2004 | 7.4 | 1.5; 13.5 | Increasing |
| | | | 2004–2015 | −4.7 | −5.5; −3.8 | Decreasing |
| | | | 2015–2017 | 1.5 | −9.2; 13.5 | Stationary |
| | | | 2001–2017 | −1.8 | −1.8; −3.3 | Decreasing |
| **SVI Human Capital** | **Number of municipalities** | | **Period** | **APC** | **(95% CI)** | **Classification** |
| Very low | 125 | 16.77 | 2001–2017 | 1.4 | 0.1; 2.7 | Increasing |

(*Continued*)

**Table 4.** (Continued)

| | | | | | | |
|---|---|---|---|---|---|---|
| Low | 1,137 | 23.20 | 2001–2017 | 0.4 | −0.3; 1.2 | Stationary |
| Medium | 1,452 | 25.64 | 2001–2003 | 10.0 | 1.0; 19.7 | Increasing |
| | | | 2003–2017 | −1.2 | −1.2; −1.6 | Decreasing |
| | | | 2001–2017 | 0.2 | −0.9; 1.2 | Stationary |
| High | 1,139 | 26.37 | 2001–2003 | 14.7 | −1.8; 34.0 | Stationary |
| | | | 2003–2017 | −3.3 | −4.0; −2.6 | Decreasing |
| | | | 2001–2017 | −3.0 | −3.0; 0.6 | Stationary |
| Very high | 1,711 | 29.22 | 2001–2003 | 12.5 | 2.0; 24.1 | Increasing |
| | | | 2003–2015 | −4.1 | −4.7; −3.5 | Decreasing |
| | | | 2015–2015 | 0.9 | −8.5; 11.3 | Stationary |
| | | | 2001–2017 | −1.5 | −3.1; 0.1 | Stationary |

APC- Annual Percent Change; CI- Confidence interval.

experienced by the population result in a heterogeneous temporal and spatial pattern of TB, with high-risk spatial-temporal clusters. It is, thus, not likely that a generic set of actions would have a favorable impact of the same intensity in all regions of the country. In the North, for example, which had the highest incidence rate and a shy percentage drop of 0.8% per year, social vulnerability is more intense than in the other regions, especially with regard to urban infrastructure and human capital, as observed in this study, in the distribution of the SVI, and in another Brazilian study.

The distribution of spatio-temporal clusters showed large areas of maintenance of the TB transmission chain, not respecting state geographical limits, as in cluster 8, (162 municipalities from three states—São Paulo, Santa Catarina, and Paraná) and cluster 10 (117 municipalities from two states—Maranhão and Piauí). Besides showing the wide dispersion of the disease in space, it reflects the need for integrated public policies among the states of the federation [13].

The spatial divergences observed between Moran's statistics and the spatial-temporal scan statistic reflect the instability of epidemiological indicators and the possible influence of the quality of disease surveillance systems. Differences were not observed in the richest regions of the country (Southeast and South). In this case, the spatiotemporal scan seems to be more adequate to understand the dynamics of the phenomenon in which variations occur between years, unlike Moran's statistic, which considers a photograph of the moment. Spatial differences in the distribution of tuberculosis have been shown in other regional and local studies, with strong influence of the social [12, 13, 22, 23]. The municipalities with greater SVI showed higher incidence rates, with the following two subindices standing out: i) Human Capital and ii) Income and Work. While Human Capital involves health conditions and access to education (two important social determinants of TB), the Income and Work subindex expresses the degree of income insufficiency and factors associated families' income security, such as adult unemployment, informal employment, family dependency, and child labor [11].

Low level of schooling and low income are factors that have multiple effects on the transmission dynamic of TB [12]. The sociodemographic profile of poverty, which has already been associated with this disease [14], limits access to food and the adoption of healthy habits. From the biological point of view, it makes access to quality food difficult, which results in compromised immune status, making individuals more susceptible to illness. This relationship is, nonetheless, bidirectional, given that the clinical condition of TB also results in secondary malnutrition [24, 25]. People with moderate to intense malnutrition (BMI $< 17.0$ kg/m$^2$) are approximately 1.8 more likely to die early (during the first four weeks of diagnosis) than people

with BMI $\geq 17.0$ kg/m$^2$ [26]. It is worth underscoring that people experiencing homelessness, people living with HIV, and people deprived of their liberty have malnutrition as part of their general condition, and these population groups are 56, 25, and 28 times more likely to develop the active form of TB, respectively [27].

Low level of schooling and exposure to poverty are factors that complicate access to services for diagnosis and treatment of the disease [23, 28, 29]. If, on one hand, exposure to social vulnerability keeps the transmission chain active in the community, it can, on the other hand, prevent patients with the disease from being diagnosed. Accordingly, we argue that social vulnerability both elevates the risk of illness and places subjects who are ill in a situation of invisibility to health services.

Public policies that guarantee better social conditions, such as increased income for those who are most vulnerable to poverty, a higher proportion of income from work, and a higher degree of formalization in the labor market, as we have found in this study, can contribute to the interruption of TB transmission. In Brazil, income transfer policies, such as *Bolsa Família*, can have favorable impacts on the process of fighting TB. We strongly recommend investigations into the effects of these policies.

For this reason, public policies must simultaneously act on social vulnerability and guarantee a universal, free, and quality health system. There are diverse benefits of strengthening public health and national programs dedicated to ending TB, whose activities should include the diagnosis and treatment of the disease, with inclusion of all affected individuals. Of the estimated 10 million cases in 2018, only 70% were detected and treated by health systems and recorded in official information systems. Brazil is not part of the list of countries with the largest gap, and, together with China, Russia, and Zimbabwe, Brazil achieved a treatment coverage rate of over 80% [1]. It is likely that these good results are due to the availability of free and universal health care to the Brazilian population through the Unified Health System (SUS, acronym in Portuguese).

In Brazil, the expansion of the primary care network in the setting of the SUS stands out as one of the most relevant aspects that place the country in a prominent position in the global fight against TB. Since 2000, the population coverage by the Family Health Strategy and Basic Care teams has increased, reaching 59.9% and 63%, respectively, in 2015. The performance of the Family Health Strategy stands out mainly in the Northeast, where it has reached 76% coverage, and it has been growing in the Southeast, where it reached 49.2%, in 2015. This expansion broadened the population's access, especially among those with lower levels of income and schooling [30]. In addition to making access to diagnosis and medications universal, the following stand out: directly observed treatment, decentralization of reference units, introduction and updating of the rapid molecular test, tests for HIV at the time of diagnosis, active search for contacts, and treatment of latent TB in populations at risk.

Nevertheless, there are still many weak points in this system, such as inequality in the quality and flow of services in Brazilian municipalities and the low degree of access of populations who have to travel to large centers to receive diagnosis, due to the slow and complex process of decentralizing the health network. Among indicators of successful treatment, the cure rates (71% of new and retracted cases; 51% of cases of TB-HIV coinfection; 61% of multidrug-resistant cases, and 41% of extensively resistant cases) are far below the goal of greater than or equal to 90% [1], which reveals a major problem and concern, given that there is an increasing tendency toward the emergence of resistant cases.

It is already known that, in order to eliminate this disease, it is necessary for there to be treatment coverage of at least 90% of the latent form, with contact testing and the use of rapid tests at the same proportion [31]. In conjunction with this, prevention is a fundamental strategy for the elimination of TB. Without preventing the reactivation of latent TB and without

the existence of a pre- and post-exposure vaccine, it will be difficult to reach global objectives [32].

With respect to the diagnostic tests proposed by the WHO, in Brazil, only 135 municipalities (with 237 pieces of equipment) conduct the rapid molecular test for tuberculosis (RMT-TB). The use of this tool is essential to diagnosis, due to its short time to results, in conjunction with information regarding whether or not the case is resistant to the main drug used in the therapeutic regimen, rifampicin. An update to the RMT-TB cartridge, the Ultra cartridge, already exists, and it promises to be more accessible with shorter time to results. This update intensifies the fight against TB, but access to this tool throughout the territory of Brazil remains limited [4].

Nevertheless, diagnosis and treatment of patients alone are not sufficient to reduce the disease burden in the country; multi-sector policies must also consider the influence of risk factors, such as diabetes mellitus, malnutrition, tobacco use, alcohol consumption, and HIV [33]. These are the five main predisposing factors, and they can contribute to worse outcomes in treatment of TB. In 2018, in Brazil, approximately 10,000 cases of TB were attributed to alcoholism, 10,000 to HIV, 8,000 to tobacco use, and 5,000 to diabetes [1]. The odds ratio of developing TB is 2.44 to 8.33 times higher in patients with diabetes mellitus than in those without the disease. For tobacco use, it is approximately 5.39 times higher (95% CI: 2.44 to 11.91), and, for alcoholism, it is 3.50 times higher (95% CI: 2.01 to 5.93) [33]. Accordingly, in order to reduce the disease burden, it is necessary to be aware of these comorbidities in patients with TB, in order to guarantee adequate management of both conditions.

It is evident that the population's living conditions and the access to health services are directly influenced by the socioeconomic and political circumstances of the country. The Brazilian political crisis that began in 2013 and culminated with the impeachment of the country's president in 2016 generated negative impacts, most of all for the poorest populations of the country, with attacks on labor rights, unemployment, reduced purchasing power of the population, and reduced investments in important social areas, such as health and education [34, 35]. We argue that this context has resulted in an increase in the incidence of TB in the years 2016 and 2017, mainly in municipalities with very high social vulnerability, given that they are the most sensitive to changes in the political and economic circumstances.

Accordingly, in order for the elimination of TB as a public health problem to stop being a global illusion and to become reality, action is necessary in different lines of work that involve the following [8, 36]: i) universal access to timely and quality diagnosis, with inclusion of all patients; ii) monitoring and follow-up of cases that are resistant to conventional treatment; iii) consolidation of prevention actions, such as expanding the coverage of the bacillus Calmette-Guérin vaccine, preventive treatment for people living with HIV, and testing of contacts of TB cases; iv) broad investment in prevention, diagnosis, and treatment actions; v) universalization of health systems and inter-sector actions; and vi) action on social determinants and risk factors for TB during the four stages of pathogenesis: exposure, infection, progression among exposed people, and health care [37].

Even considering the methodological precautions adopted, this study does have some limitations, including the quality of data registered in the SINAN. The records are influenced by the quality of the health surveillance services of Brazilian states and municipalities, which, in the North and Northeast Regions, often face severe problems, mainly related to the qualification of human resources and the lack of working conditions, which result in underreporting of cases. In this study, 22% of municipalities (n = 1,216) would meet the criterion for elimination of TB (< 10 cases per 100,000); after smoothing of the rates, however, this number changed to scarcely over 5% (n = 305). This information is extremely useful in confirming the need to empower epidemiological surveillance and make it more effective, as well as the indispensable investigation of contacts, given that TB is an airborne disease.

## Conclusion

Brazil reduced the incidence of TB from 42.8 per 100,000 inhabitants to 35.2 per 100,000 between 2001 and 2017. All five major regions also showed a decreasing trend, and the Northeast stood out with the greatest percentage decrease. With respect to Brazilian states, only Minas Gerais showed an increasing trend, and nine states showed a stationary trend.

A total of 326 Brazilian municipalities were classified as high priority from the epidemiological and statistical point of view for political and administrative intervention in the fight against TB. Moreover, 22 high-risk spatial-temporal clusters were identified, involving a total of 561 municipalities. Finally, it was found that the epidemiological situation of nearly one thousand municipalities is underreported, and these municipalities must have more cases than are officially registered.

The overall SVI and the subindices of Human Capital and Work and Income were associated with the incidence of TB; they also showed an association with other disaggregate variables. Furthermore, it was observed that the incidence rates were higher in municipalities with greater social vulnerability, confirming its status as a disease of neglected populations.

TB is a global health and political issue, and Brazil has the means necessary to fulfill the goals and become a leader in the international scenario of fighting the disease. Accordingly, the results of this study contribute to understanding the spatial distribution of TB in Brazil, highlighting the importance of spatial analysis associated with socioeconomic indicators as methodological tools to assist in planning, execution, and assessment of health actions, thus guiding interventions for the elimination of the disease.

## Supporting information

**S1 Table. Performance of these two models—OLS and SLM.**
(DOCX)

**S2 Table. Incidence rate of tuberculosis per 100,000 inhabitants by Social Vulnerability Index classification and subindices.** Brazil, 2001–2017.
(DOCX)

## Author Contributions

**Conceptualization:** João Paulo Silva de Paiva, Mônica Avelar Figueiredo Mafra Magalhães, Thiago Cavalcanti Leal, Leonardo Feitosa da Silva, Lucas Gomes da Silva, Rodrigo Feliciano do Carmo, Carlos Dornels Freire de Souza.

**Data curation:** João Paulo Silva de Paiva, Mônica Avelar Figueiredo Mafra Magalhães, Thiago Cavalcanti Leal, Lucas Gomes da Silva, Rodrigo Feliciano do Carmo, Carlos Dornels Freire de Souza.

**Formal analysis:** João Paulo Silva de Paiva, Mônica Avelar Figueiredo Mafra Magalhães, Thiago Cavalcanti Leal, Lucas Gomes da Silva, Rodrigo Feliciano do Carmo, Carlos Dornels Freire de Souza.

**Funding acquisition:** João Paulo Silva de Paiva, Mônica Avelar Figueiredo Mafra Magalhães, Leonardo Feitosa da Silva, Lucas Gomes da Silva, Rodrigo Feliciano do Carmo, Carlos Dornels Freire de Souza.

**Investigation:** João Paulo Silva de Paiva, Thiago Cavalcanti Leal, Lucas Gomes da Silva, Rodrigo Feliciano do Carmo, Carlos Dornels Freire de Souza.

**Methodology:** João Paulo Silva de Paiva, Lucas Gomes da Silva, Rodrigo Feliciano do Carmo, Carlos Dornels Freire de Souza.

**Project administration:** João Paulo Silva de Paiva, Thiago Cavalcanti Leal, Leonardo Feitosa da Silva, Lucas Gomes da Silva, Rodrigo Feliciano do Carmo, Carlos Dornels Freire de Souza.

**Resources:** João Paulo Silva de Paiva, Mônica Avelar Figueiredo Mafra Magalhães, Thiago Cavalcanti Leal, Leonardo Feitosa da Silva, Lucas Gomes da Silva, Rodrigo Feliciano do Carmo, Carlos Dornels Freire de Souza.

**Software:** João Paulo Silva de Paiva, Thiago Cavalcanti Leal, Leonardo Feitosa da Silva, Lucas Gomes da Silva, Rodrigo Feliciano do Carmo, Carlos Dornels Freire de Souza.

**Supervision:** João Paulo Silva de Paiva, Mônica Avelar Figueiredo Mafra Magalhães, Thiago Cavalcanti Leal, Lucas Gomes da Silva, Rodrigo Feliciano do Carmo, Carlos Dornels Freire de Souza.

**Validation:** João Paulo Silva de Paiva, Leonardo Feitosa da Silva, Lucas Gomes da Silva, Rodrigo Feliciano do Carmo, Carlos Dornels Freire de Souza.

**Visualization:** João Paulo Silva de Paiva, Leonardo Feitosa da Silva, Lucas Gomes da Silva, Rodrigo Feliciano do Carmo, Carlos Dornels Freire de Souza.

**Writing – original draft:** João Paulo Silva de Paiva, Mônica Avelar Figueiredo Mafra Magalhães, Lucas Gomes da Silva, Rodrigo Feliciano do Carmo, Carlos Dornels Freire de Souza.

**Writing – review & editing:** João Paulo Silva de Paiva, Mônica Avelar Figueiredo Mafra Magalhães, Thiago Cavalcanti Leal, Leonardo Feitosa da Silva, Lucas Gomes da Silva, Rodrigo Feliciano do Carmo, Carlos Dornels Freire de Souza.

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
