## [Decision Letter · Decision Letter 0]

24 Sep 2021

PONE-D-21-04812

Time Trend, Social Vulnerability, and Identification of Risk Areas for Tuberculosis in Brazil: an Ecological Study

PLOS ONE

Dear Dr. Souza,

Thank you for submitting your manuscript to PLOS ONE. After careful consideration, we feel that it has merit but does not fully meet PLOS ONE’s publication criteria as it currently stands. Therefore, we invite you to submit a revised version of the manuscript that addresses the points raised during the review process.

The reviewers have highlighted concerns regarding the spatial analysis results, model validation, and analysis techniques which require addressing for this manuscript to meet our publication criteria.

In particular, for our criteria that the 'Experiments, statistics, and other analyses are performed to a high technical standard and are described in sufficient detail' https://journals.plos.org/plosone/s/criteria-for-publication#loc-3, please address Reviewer 1's comments regarding: space-time scan statistics and Moran's I, LISA comparisons and weight matrix settings, edge effect, explanatory variables, differential results and comparisons between spatial lag and OLS, and model validation. Please also address Reviewer 2's comments regarding space-time scan statistics

To meet our publication criteria that 'Conclusions are presented in an appropriate fashion and are supported by the data' https://journals.plos.org/plosone/s/criteria-for-publication#loc-4, please address Reviewer 2's comments regarding the discussion of the spatial analysis results.

We look forward to receiving your revised manuscript.

Kind regards,

Sebastian Shepherd

Associate Editor

PLOS ONE

Journal Requirements:

2. If available, please provide a table of participant demographics.

3. We note that Figures 1,2,3 and 4 in your submission contain map images which may be copyrighted. All PLOS content is published under the Creative Commons Attribution License (CC BY 4.0), which means that the manuscript, images, and Supporting Information files will be freely available online, and any third party is permitted to access, download, copy, distribute, and use these materials in any way, even commercially, with proper attribution. For these reasons, we cannot publish previously copyrighted maps or satellite images created using proprietary data, such as Google software (Google Maps, Street View, and Earth). For more information, see our copyright guidelines: http://journals.plos.org/plosone/s/licenses-and-copyright.

a. You may seek permission from the original copyright holder of Figures 1,2,3 and 4 to publish the content specifically under the CC BY 4.0 license.  

Natural Earth (public domain): http://www.naturalearthdata.com/.

Reviewers' comments:

Reviewer's Responses to Questions

**Comments to the Author**

1. Is the manuscript technically sound, and do the data support the conclusions?

Reviewer #1: Yes

Reviewer #2: Partly

2. Has the statistical analysis been performed appropriately and rigorously? 

Reviewer #1: Yes

Reviewer #2: No

3. Have the authors made all data underlying the findings in their manuscript fully available?

Reviewer #1: No

Reviewer #2: Yes

4. Is the manuscript presented in an intelligible fashion and written in standard English?

Reviewer #1: Yes

Reviewer #2: Yes

5. Review Comments to the Author

Reviewer #1: The authors mainly evaluate the spatial and temporal distribution of TB in Brazil during the period from 2001 to 2017.

0. While I appreciate authors detailed explanations, in my opinion the manuscript is very lengthy, and some explanations are either redundant or can be moved to the supplementary materials. The authors need to shorten the lengths of manuscript to at least half of current length while maintaining the quality of work.

1. The explanations for space-time scan statistics doesn’t contain time component and is more like pure spatial scan statistics. If yes, what’s the difference between Local Moran’s I and this method?

2. Authors need to compare the results of LISA and spatial scan statistics.

3. Explain the settings of weight matrix in LISA such as weight matrix, or settings of space-time scan statistics.

4. How did you address edge effect in the cluster detection techniques.

5. Authors need to justify why they included these explanatory variables in the study.

6. According to Table 7 there are significant differences between the results of spatial lag and OLS which is unusual and makes the results unreliable. Also the authors need to compare the performance of these two models.

7. Modeling without validation is not reliable. The authors need to test the performance of models on an independent (test) dataset.

Reviewer #2: 1. In this paper the spatiotemporal scan statistic has been used and identified spatiotemporal clusters as in Table 2. However, the authors have used the term “spatial clusters” throughout the paper e.g., in Abstract: “22 high-risk spatial clusters were identified”. Also in Figure 4, the title is “Spatial-temporal scan statistics of cases of tuberculosis. Brazil, 2001-2017”, but it doesn’t show time-frame for each cluster according to Table 2.

It seems the aim is to identify spatial clusters. The spatial-temporal scan statistic is invalid for spatial analysis. The spatial scan statistic should be used for spatial analysis.

2. In the discussion section, the results are not specifically discussed e.g., spatial analysis results (moran’s I and Spatial scan statistic) have not been discussed properly. This section includes some details irrelevant to the results (e.g., first 5 paragraphs), These details should be in the introduction section.

6. PLOS authors have the option to publish the peer review history of their article (what does this mean?). If published, this will include your full peer review and any attached files.

Reviewer #1: No

Reviewer #2: **Yes: **Sami Ullah

---

## [Author Response · Author response to Decision Letter 0]

17 Oct 2021

Response to Reviewers

PONE-D-21-04812- Time Trend, Social Vulnerability, and Identification of Risk Areas for Tuberculosis in Brazil: an Ecological Study

Dear Editor,

We appreciate very much for the kind consideration of our manuscript for publication. We thank the reviewers for their kind comments. We have tried to reply to the comments of the reviewers to our best as shown underneath and also included in the text. We hope that our manuscript is now fit for publication.

Thanking you in advance.

EDITOR

In particular, for our criteria that the 'Experiments, statistics, and other analyses are performed to a high technical standard and are described in sufficient detail' https://journals.plos.org/plosone/s/criteria-for-publication#loc-3, please address Reviewer 1's comments regarding: space-time scan statistics and Moran's I, LISA comparisons and weight matrix settings, edge effect, explanatory variables, differential results and comparisons between spatial lag and OLS. Please also address Reviewer 2's comments regarding space-time scan statistics.

ANSWER: We have corrected all recommendations: 1- expansion of the methods section, with details of the spatial and space-time analyses performed; 2- inclusion of supplementary table 1 with the performance of the two models; 3- correction of the use of the term space-time. Please see details in response to reviewers.

To our publication criteria that 'Conclusions are presented in an appropriate fashion and are supported by the data' https://journals.plos.org/plosone/s/criteria-for-publication#loc-4, please address Reviewer 2's comments regarding the discussion of the spatial analysis results.

ANSWER: We have adjusted the discussion as recommended by reviewer 2.

Journal Requirements:

ANSWER: Accordingly.

2. If available, please provide a table of participant demographics.

ANSWER: Demographic data are published by the Brazilian Ministry of Health in periodical bulletins. In addition, the data have variations between collection times. At the time the research was carried out, the database had 1 243 629 cases. On verification today, October 15, 2020, the database has 1 243 501 (a difference of 128 records). This is due to systematic, daily adjustments in the bank. For this reason, we have chosen not to add new data.

3. We note that Figures 1,2,3 and 4 in your submission contain map images which may be copyrighted. All PLOS content is published under the Creative Commons Attribution License (CC BY 4.0), which means that the manuscript, images, and Supporting Information files will be freely available online, and any third party is permitted to access, download, copy, distribute, and use these materials in any way, even commercially, with proper attribution. For these reasons, we cannot publish previously copyrighted maps or satellite images created using proprietary data, such as Google software (Google Maps, Street View, and Earth). For more information, see our copyright guidelines: http://journals.plos.org/plosone/s/licenses-and-copyright.

a. You may seek permission from the original copyright holder of Figures 1,2,3 and 4 to publish the content specifically under the CC BY 4.0 license. 

Natural Earth (public domain): http://www.naturalearthdata.com/.

ANSWER: The map meshes used here were obtained from Natural Earth (public domain) <http://www.naturalearthdata.com/about/terms-of-use/> covered by a Creative Commons Attribution 4.0 International (CCBY) License (https://creativecommons.org/licenses/by/4.0/legalcode). The base layers of the maps were modified in QGIS software version 2.18.

Review Comments to the Author

Reviewer #1: 

The authors mainly evaluate the spatial and temporal distribution of TB in Brazil during the period from 2001 to 2017.

0. While I appreciate authors detailed explanations, in my opinion the manuscript is very lengthy, and some explanations are either redundant or can be moved to the supplementary materials. The authors need to shorten the lengths of manuscript to at least half of current length while maintaining the quality of work.

ANSWER: We appreciate the reviewer's suggestion. We have adjusted the text to make it clearer for the reader. In addition, we have added a supplementary material to avoid increasing the length of the manuscript. We hope that the text looks better in this new version.

1. The explanations for space-time scan statistics doesn’t contain time component and is more like pure spatial scan statistics. If yes, what’s the difference between Local Moran’s I and this method?

ANSWER: We thank the reviewer for the comment. We performed spatial-temporal analysis and Local Moran's I. In the text, we have added a detailed explanation of the spatial-temporal analysis. Thus, the analyses were Moran's I and spatial-temporal scanning. The local Moran's Index does not have the ability to specify the exact location of each specific cluster nor its relative risk in relation to the others. Thus, the spatial-temporal scan complements the analysis and understanding of the phenomenon. 

In the SatScan Manual, the time-space scan is defined: “When the scan statistic is used to evaluate the spatial variation in temporal trends, the scanning window is purely spatial in nature. The temporal trend is then calculated inside as well as outside the scanning window, for each location and size of that window. The null hypothesis is that the trends are the same, while the alternative is that they are different. Based on these hypotheses, a likelihood is calculated, which is higher the more unlikely it is that the difference in trends is due to chance. The most likely cluster is the cluster for which the temporal trend inside the window is least likely to be the same as the temporal trend outside the cluster. This could be because of various reasons. For example, if the temporal trend inside the cluster is higher, it could be because all areas has the same incidence rate of a disease at the beginning of the time period, but the cluster area has a higher rate at the end of the time period. It could also be because the cluster area has a lower incidence rate at the beginning of the time period, after which it ‘catches up’ with the rest so that the rate is about the same at the end of the time period. Hence, a statistically significant cluster in the spatial variation in temporal trend analysis does not necessarily mean that the overall rate of disease is higher or lower in the cluster. The spatial variation in temporal trends scan statistic can only be run with the discrete Poisson probability model. For it to work, it is important that the total study period length is evenly divisible by the length of the time interval aggregation, so that all time intervals have the same number of years, if it is specified in years, the same number of months if it is specified in months or the same number of days if it is specified in days.”

2. Authors need to compare the results of LISA and spatial scan statistics.

ANSWER: We have added a paragraph in the results section making this comparison, as recommended by the reviewer.

3. Explain the settings of weight matrix in LISA such as weight matrix, or settings of space-time scan statistics.

ANSWER: For Moran's analysis, we added the explanation, " The spatial weight matrix adopted in this study was contiguity or adjacency matrix [Wij=1], if [Ai] shares a common side with [Aj]; otherwise [Wij=0], where [W] refers to Weight, and each value [Wij] depends on the spatial relationship between locations [i] and [j]. In these cases, municipalities share a border (first-order neighbors). All objects that are nearby were considered to have the same weight.”

The settings of the spatial-temporal scan were explained in the corresponding session: “The scan statistic establishes a flexible circular window in the map, positioned on each of several centroids, whose radius is established in 50% of the total population at risk, with a maximum radius of 500 km. The flexibility of the window is justified by the fact that the size of the clusters are not known a priori, given that the population at risk is not geographically homogeneous. Monte Carlo simulations (adopting 999 permutations) were used to obtain p values, and clusters with p value < 0.05 were considered significant. Subsequently, clusters with two or fewer municipalities were excluded”.

4. How did you address edge effect in the cluster detection techniques.

ANSWER: Considering that the municipal spatial units are already delimited by objective criteria of the Brazilian Institute of Geography and Statistics (IBGE), in this work the edge effect was not considered in the results.

5. Authors need to justify why they included these explanatory variables in the study.

ANSWER: We selected 34 social indicators that express living conditions and social vulnerability of the Brazilian population that may be associated with the maintenance of the TB transmission chain in the Brazilian territory, as the literature has pointed out. We emphasize that the indicators are available for all Brazilian municipalities (national coverage) and collected with technical quality by official institutions in the country. We have added in the methods section this explanation about the indicators in order to make the text clearer for readers.

6. According to Table 3 there are significant differences between the results of spatial lag and OLS which is unusual and makes the results unreliable. Also the authors need to compare the performance of these two models.

ANSWER: We adopted the model proposed by Luc Anselim, in which the OLS model is first tested. Then, the regression residuals are submitted to Moran statistics to evaluate the effect of space on the model. If Moran's statistics indicate the existence of spatial dependence, it is recommended to use Lagrange multiplier tests to define which spatial model is the most appropriate: whether Spatial Error Model or Spatial Lag Model. Finally, the models are compared using the Akaike (AIC), Schwarz Bayesian (BIC), R2, log likelihood, and the Moran index statistic of the residuals. 

We considered as the best model the one whose AIC and BIC values were lower, the Log likelihood and R2 were higher, and the residues showed spatial independence. We compared the models and the results showed that the Spatial Lag Model (SLM) performed better. Initially, we did not add in the text of the manuscript due to space limitations. However, a supplementary table with comparative model data has been added (supplementary 1).

Reviewer #2: 

1. In this paper the spatiotemporal scan statistic has been used and identified spatiotemporal clusters as in Table 2. However, the authors have used the term “spatial clusters” throughout the paper e.g., in Abstract: “22 high-risk spatial clusters were identified”. Also in Figure 4, the title is “Spatial-temporal scan statistics of cases of tuberculosis. Brazil, 2001-2017”, but it doesn’t show time-frame for each cluster according to Table 2.

ANSWER: We thank you for the comment. We have adjusted the text and figure 4.

It seems the aim is to identify spatial clusters. The spatial-temporal scan statistic is invalid for spatial analysis. The spatial scan statistic should be used for spatial analysis.

We performed temporal, spatial and spatial-temporal analysis. We adjusted the aim of the study and added an explanation in the methods section, as also recommended by reviewer 1.

2. In the discussion section, the results are not specifically discussed e.g., spatial analysis results (moran’s I and Spatial scan statistic) have not been discussed properly. This section includes some details irrelevant to the results (e.g., first 5 paragraphs), These details should be in the introduction section.

RESPONSE: We agree that there is too much text in those first paragraphs. For that reason, we have moved some of the text to the introduction and eliminated unnecessary parts. We have also added a discussion of the specific spatial analysis addressing the differences between Moran's and spatio-temporal scan results.

---

## [Decision Letter · Decision Letter 1]

22 Dec 2021

Time Trend, Social Vulnerability, and Identification of Risk Areas for Tuberculosis in Brazil: an Ecological Study

PONE-D-21-04812R1

Dear Dr. Souza,

We’re pleased to inform you that your manuscript has been judged scientifically suitable for publication and will be formally accepted for publication once it meets all outstanding technical requirements.

Kind regards,

Michele Tizzoni

Academic Editor

PLOS ONE

Additional Editor Comments (optional):

I personally assessed the revised version of the manuscript in light of the comments made by Referee #1 and I believe this version adequately addresses the main issues raised by the Referee regarding the spatial analysis.  

Reviewers' comments:

Reviewer's Responses to Questions

**Comments to the Author**

1. If the authors have adequately addressed your comments raised in a previous round of review and you feel that this manuscript is now acceptable for publication, you may indicate that here to bypass the “Comments to the Author” section, enter your conflict of interest statement in the “Confidential to Editor” section, and submit your "Accept" recommendation.

Reviewer #2: All comments have been addressed

2. Is the manuscript technically sound, and do the data support the conclusions?

Reviewer #2: Yes

3. Has the statistical analysis been performed appropriately and rigorously? 

Reviewer #2: Yes

4. Have the authors made all data underlying the findings in their manuscript fully available?

Reviewer #2: Yes

5. Is the manuscript presented in an intelligible fashion and written in standard English?

Reviewer #2: Yes

6. Review Comments to the Author

Reviewer #2: The Introduction and Discussion sections are still very lengthy: comtains unnecessery explanations. it need to be shortened.

7. PLOS authors have the option to publish the peer review history of their article (what does this mean?). If published, this will include your full peer review and any attached files.

Reviewer #2: **Yes: **Sami Ullah

---

## [Editor Report · Acceptance letter]

14 Jan 2022

PONE-D-21-04812R1 

Time Trend, Social Vulnerability, and Identification of Risk Areas for Tuberculosis in Brazil: an Ecological Study 

Dear Dr. de Souza:

I'm pleased to inform you that your manuscript has been deemed suitable for publication in PLOS ONE. Congratulations! Your manuscript is now with our production department. 

Kind regards, 

on behalf of

Dr. Michele Tizzoni 

Academic Editor

PLOS ONE